# Active Tactile Sensibility in Implant Prosthesis vs. Complete Dentures: A Psychophysical Study

**DOI:** 10.3390/jcm11226819

**Published:** 2022-11-18

**Authors:** Diego González-Gil, Ibrahim Dib-Zaitun, Javier Flores-Fraile, Joaquín López-Marcos

**Affiliations:** Surgery Department, Dental Clinic Faculty of Medicine, University of Salamanca, 37007 Salamanca, Spain

**Keywords:** osseoperception, tactile sensibility, interocclusal thickness, interocclusal perception

## Abstract

Background and Objectives: Proprioceptive information from natural dentition and adjacent oral tissues enables correct masticatory function, avoiding damage to the teeth. Periodontium is the main source of this relevant information, and when a tooth is lost, all this proprioceptive sensibility relies on receptors from muscles, the mucous membrane or the temporomandibular joint, and this sensibility gets worse. Active tactile sensibility measures this proprioceptive capability in microns by psychophysical studies consisting of introducing thin metal foils between patients’ dental arches during chewing to see if they are able to notice them or not. Osseoperception is a complex phenomenon that seems to improve this sensibility in patients wearing dental implants. The objective of this investigation is to measure this sensibility in different prosthetic situations by performing a psychophysical investigation. Material and Methods: We divided 67 patients in three groups depending on their prosthetic situation and performed a psychophysical study by introducing aluminium foils of different thicknesses in order to establish an active tactile sensibility threshold in every group. We also measured variables such as prosthetic wearing time, age or gender to see how they may influence threshold values. We used Student’s t-test and Mann–Whitney U tests to analyse these results. Results: Active tactile sensibility threshold values in implants are lower than those from complete dentures but higher than values in natural dentition. However, values in implants are closer to natural dentition than complete denture values. Age, gender or prosthetic wearing time have no influence in active tactile sensibility thresholds. Conclusion: Active tactile sensibility threshold values depend on prosthetic rehabilitations and the mechanoreceptors involved in every situation. Implant prosthesis presents an increased active tactile sensibility thanks to osseoperception phenomenon.

## 1. Introduction

Proprioceptive mechanoreceptors provide a lot of information to the brain cortex in order to perform masticatory function in a proper way [1]. The main source of these receptors is located in the periodontal ligament [2,3,4,5,6,7,8]. Additionally, there are proprioceptive receptors in adjacent tissues, such as masticatory muscles, mucous membranes or the temporomandibular joint; however, these are not as accurate as those from periodontium [9,10,11,12]. For instance, these receptors inform about the hardness of food or the force used while chewing, preventing us from harming our teeth [13,14,15,16].

When a tooth is extracted, the periodontium gets lost, as well; as well as this important information, so useful during masticatory function. When a patient is fully edentulous and wears a complete denture, there is also a lack of sensitive information due to the absence of periodontium. Then, receptors from adjacent tissues must embrace all the proprioceptive information during chewing, and the masticatory function gets worse [17,18,19,20].

This proprioceptive information seems to be improved in patients rehabilitated with implant prosthesis, thanks to a phenomenon called osseoperception [21,22,23,24,25,26,27,28]. This phenomenon is not well known yet, but it seems to be associated to mechanoreceptors from peri-implant tissues. Its definition consists of the sensation arising from the mechanical stimulation of implant prosthesis after the activation of receptors from peri-implant bone [29,30]. The information obtained by receptors from adjacent tissues, together with osseoperception, allows these patients to present a more increased proprioceptive sensibility than those rehabilitated with complete dentures [31,32,33]. Corpas investigation [34] has helped to understand this phenomenon by discovering the presence of nerve fibres around peri-implant bone surrounding dental implants extracted due to mechanical failure.

After the rehabilitation with dental implants, there is not only a change in oral afferent pathways but also an adjustment in the primary somatosensorial cortex [35,36,37,38]. The influence of these rehabilitations in the brain cortex has been studied by Habre-Hallage [39,40,41], who used fMRI (functional magnetic resonance imaging) while stimulating implant abutments with specific devices in order to see how the cortex is activated. These kinds of studies are neurophysiological investigations and are very scarce in literature, as they are less relevant to our dental office routine. On the other hand, psychophysical investigations are more useful as they are easier to reproduce in our offices and provide information about afferent pathways and their relationship with prosthetic rehabilitations. These studies are focused on tactile sensibility, which is used to evaluate proprioception of mechanoreceptors from the masticatory system [25,26,42,43,44].

There are two kinds of tactile sensibility depending on the receptors activated during the psychophysical studies, and they are measured in different values. Passive tactile sensibility is measured by Newtons, as it consists of the minimum force a patient can perceive when a device, designed to apply pressure at different intensities and directions, stimulates a tooth or an implant abutment [45,46]. In this case, only receptors from periodontium or peri-implant tissues are activated. Since receptors from adjacent tissues are not stimulated, this sensibility cannot be measured in complete denture wearers. This fact, along with the complexity of designing these devices, makes these investigations very scarce in literature. Active tactile sensibility is measured by micrometres, and its thresholds are evaluated by introducing thin metal foils between teeth during chewing to check if patients are able to perceive them or not [47,48,49,50,51,52,53]. These studies are easier to perform as they are very similar to those we make during occlusal adjustment in our prosthesis. Besides that, the forces received by the teeth in these investigations are better reproduced with natural mandibular movements. Finally, active tactile sensibility studies involve all kinds of oral mechanoreceptors stimulated during masticatory function, so it can be measured in patients rehabilitated with complete dentures, as well [54,55,56,57].

After performing psychophysical investigations, it is possible to obtain threshold values of every patient and group of study in order to be able to compare these results. Specifically, it is really useful to contrast mean values between groups to see if there are any differences in tactile sensibility. Thus, every prosthetic situation can be measured in terms of active tactile sensibility, as well as which mechanoreceptors are stimulated in each group and their influences in threshold values [58,59,60]. The purpose of this investigation is to perform a psychophysical study about active tactile sensibility in complete denture wearers and patients rehabilitated with dental implants. This procedure will allow us to obtain sensibility values of each kind of prosthesis in order to compare them and check which one presents an increased sensibility and more natural functioning. The null hypothesis is that prosthetic rehabilitations do not influence active tactile sensibility values. The alternative hypothesis assumes that implant prosthesis presents an increased sensibility value with respect to complete dentures.

Besides that, other variables such as age, gender or prosthetic wearing time will be taken into account to see their influence in mean threshold values of tactile sensibility.

## 2. Materials and Methods

### 2.1. Study Design

A psychophysical study has been carried out to measure active tactile sensibility in three groups of patients with different prosthetic rehabilitations. In total, 67 patients with different prosthetic rehabilitations were included in this investigation.

Group A is formed by 17 patients wearing complete dentures in both arches. Group B is made up of 20 patients with implant prosthesis, whose antagonist is also an implant rehabilitation. Finally, group C is formed by 30 patients rehabilitated with implant prosthesis whose antagonist is natural dentition. Active tactile sensibility is measured in micrometres by introducing metal foils from different thicknesses while patients are chewing. The thinnest foil perceived by every patient is considered as the minimum threshold value. Mean values in each group are calculated to compare active tactile sensibility in different prosthetic situations.

In this study, aluminium foils from three thicknesses (14, 30 and 50 micrometres) were introduced between the dental arches of every patient in each group. The values of these foils have been chosen as they coincide with mean threshold values of prosthetic rehabilitations that are present in literature [60]. Methodologies and protocols of the main psychophysical studies that measure active tactile sensibility have been unified in this investigation [47,48,49,50,51,52,53,54,55,56,57,58].

The study consisted of a random sequence of twenty introductions of each aluminium foil that was carried inside the mouth by a Miller forceps. In order to avoid incorrect responses and to obtain reliable results, we designed a placebo test. This placebo test consisted of the introduction of Miller forceps inside the mouth without carrying any aluminium foil attached. In this way, we were able to check if the patient understood the procedure or if he was telling us the truth, since, during this intervention, the patient must not perceive the foil between his or her dental arches.

In addition, every patient was sitting in a comfortable position in a dental chair, blindfolded and acoustically isolated. This isolation was performed by putting headphones in the patient’s ears and playing pink noise. This procedure prevented patients from auditorily perceiving foils during chewing.

Threshold values were obtained by following the 50% rule of correct answers during the introduction of foils [48,49,50,51,54]. Every patient must perceive each foil in at least half of all introductions in order to set an active tactile sensibility threshold value. After collecting the results, mean values were calculated and statistically evaluated by using Student’s t-tests and Mann–Whitney U tests. This investigation is registered in the ISRCTN registry with number ISRCTN44091392, and the Bioethics Committee from the University of Salamanca approved its ethical clearance.

### 2.2. Inclusion Criteria

Patients from the University of Salamanca dental clinic and Diego González Gil’s private office have participated in this investigation voluntarily. Every patient has been examined in detail by investigators to make sure that they meet inclusion criteria. Only adults and healthy patients, rehabilitated with complete dentures or implant prosthesis with a proper adjustment, were included in this investigation.

### 2.3. Exlusion Criteria

Patients with maladjusted prosthesis or patients suffering from temporomandibular disorders were not included in this investigation, as well as patients unable to understand how this procedure works. Furthermore, patients that do not pass the placebo test were excluded from the study in order to obtain reliable values.

### 2.4. Variables

Four different variables were taken into account to measure active tactile sensibility in each group: type of prosthetic situation, age, gender and prosthesis wearing time, measured in years. In this way, we were able to ascertain whether these variables influenced active tactile sensibility values.

### 2.5. Resources

This investigation was performed in dental facilities from Diego González Gil’s private office and in the University of Salamanca Dental Clinic. Each one is equipped with every material needed to perform the psychophysical study.

## 3. Results

A total of 70 patients were included in this investigation, but 3 of them were excluded as a result of not passing the placebo test. The remaining 67 patients were divided into different groups depending on their prosthetic situation, as is represented in Table 1. After obtaining the minimum threshold values of every patient, measured in micrometres and related to the thinnest foil perceived by each one, mean values from three groups were calculated, and they are represented in Table 2. Then, we performed descriptive statistics of each group to obtain mean and median values of age and prosthetic wearing time, as well as their distribution according to gender. Additionally, statistical analysis by using Student’s t-test and Mann–Whitney U test was carried out to obtain *p*-values of groups A, B and C. These results are represented in Table 3.

## 4. Discussion

These results show that active tactile sensibility threshold values are increased in patients from group A with respect to those from group B. In this way, our alternative hypothesis is accepted, as implant prosthesis presents an increased active tactile sensibility with respect to complete dentures. Specifically, active tactile sensibility thresholds in complete denture wearers have twice the value of implant prosthesis wearers. These results agree with those from other authors [54,56,60] which suggest that implant prosthesis wearers had an increased active tactile sensibility over patients wearing complete dentures. This fact is due to the presence of osseoperception phenomenon in implant prosthetic rehabilitations. This phenomenon is still not fully understood. Its origin and functioning are unknown. Corpas’ investigation [34] suggests that nerve fibres surrounding peri-implant tissues may be involved in this complex phenomenon. These fibres, present in Haversian canals adjacent to dental implants, could be responsible for an increased tactile sensibility. Moreover, studies from Habre-Hallage [39,40] have demonstrated how the primary somatosensory cortex responds while implants are stimulated, as well as how neural pathways are readjusted in order to integrate implant prosthesis. 

Therefore, this phenomenon enables a better proprioceptive performance of masticatory function [25,26,29,30]. Further, the threshold values in group C are lower than those from group B, although they are very similar to each other. As patients from group C present natural teeth, proprioceptive information from the periodontium allows them to present an increased sensibility that cannot be equated with that obtained by osseoperception in implant prosthesis wearers [47,48,49,50,51,52]. These results show how implant prosthesis presents an improved tactile sensibility with respect to complete dentures. The clinical importance of this study is to prove that dental implants are a better solution to restoring edentulous patients than complete dentures, in terms of proprioception. For instance, implant prosthesis wearers can perceive changes in food hardness during chewing more easily than complete denture wearers. Moreover, these rehabilitations allow a better control of masticatory function by applying proper forces while eating. In this investigation, we have performed a psychophysical study by mixing the most reliable procedures in the scarce literature. For instance, we conducted a systematic review to study active tactile sensibility in different prosthetic situations, as we thought it could be useful before starting our psychophysical investigation. In this way, we could adapt the thicknesses of the metal foils in our study to the mean values of active tactile sensibility thresholds that we found in our review. We also used other procedures such as visual or acoustic isolation or placebo tests. The main limitation of this study is the partial ignorance about osseoperception phenomenon, as it is still impossible to completely comprehend its function. Along with this, multiple and single implant prostheses were included in group B without making a distinction between both rehabilitations. Furthermore, more studies with a larger sample size and representing more prosthetic situations can be recommended to obtain more detailed values.

The active tactile sensibility thresholds of each group match with the mean values obtained in our previous work [60]. After comparing values in both investigations, they turned out to be very similar, with an increased sensibility in implant prosthesis wearers with regard to complete denture wearers. However, this increased sensibility presents a higher threshold value than active tactile sensibility in natural dentition.

Therefore, it is clear that active tactile sensibility is influenced by the prosthetic rehabilitation that every patient presents. Additionally, every kind of prosthetic rehabilitation activates different proprioceptive receptors during masticatory function, which also influences threshold values. For instance, complete denture wearers only activate receptors from muscles, mucous membranes or the temporomandibular joint during chewing, which has a negative influence in active tactile sensibility threshold values [17,18,19,20]. Patients rehabilitated with dental implants also activate receptors from adjacent tissues but, thanks to the osseoperception phenomenon, present improved threshold values over complete denture wearers [54,56]. The lowest values of active tactile sensibility thresholds are those from natural dentition due to the presence of periodontium, which presents a lot of proprioceptive receptors that provide very relevant information during masticatory function [10,15,25,26].

There is not a consensus in literature about the influence of other variables such as gender, age or prosthetic wearing time in active tactile sensibility threshold values. While some studies claim that these variables do not influence threshold values [48,49], another denies this fact [52]; therefore, we thought it would be interesting to study these variables in our investigation. Thus, we performed a statistical analysis of the results in each group by using Student’s t-test and Mann–Whitney U tests in order to discover this potential influence.

The results of our analysis agree with those from Enkling’s study [49] in which other variables different from the prosthetic situation do not influence active tactile sensibility values. After performing statistical analysis in all groups, *p*-values were much higher than the significance value of 0.05, as is shown in Table 3.

## 5. Conclusions

(1) Implant prosthesis presents an improved active tactile sensibility with respect to complete dentures due to osseoperception; however, more studies are needed to understand how this phenomenon works.

(2) Age, gender or prosthetic wearing time have no influence in active tactile sensibility threshold values.

(3) Active tactile sensibility threshold values are influenced by the proprioceptive receptors activated during masticatory function, which depend on the kind of prosthetic rehabilitation existing in every situation.

## Figures and Tables

**Table 1 jcm-11-06819-t001:** Sample size.

	Group A	Group B	Group C
Prosthetic situation	Complete denture wearers in both arches	Implant prosthesis wearers in both arches	Implant prosthesis wearers which antagonistic are natural teeth
Number of patients included	17	20	30

**Table 2 jcm-11-06819-t002:** Mean values of active tactile sensibility thresholds.

Group	A	B	C
Mean threshold value	48.82 µm	24.4 µm	14.5 µm
Standard deviation	4.85 µm	7.83 µm	2.97 µm

**Table 3 jcm-11-06819-t003:** Group A, B and C statistics.

	Group A		Men		Women		
	Mean ± SD	Median (IR)	Media ± SD	Median (IR)	Mean ± SD	Median (IR)	*p*-Value
Age	77.59 ± 10.98	79 (58–95)	75.38 ± 12.33	75.50 (58–95)	79.56 ± 9.94	79 (66–92)	0.451
PWT	2.24 ± 2.71	2 (0–10)	2.25 ± 3.41	1 (0–10)	2.22 ± 2,11	2 (0–7)	0.541
	**Group B**		**Men**		**Women**		
	Mean ± SD	Median (IR)	Mean ± SD	Median (IR)	Mean ± SD	Median (IR)	*p*-Value
Age	59.20 ± 7.09	58 (45–72)	59.80 ± 4.20	55 (53–70)	57.40 ± 7.16	61 (45–72)	0.527
PWT	6.05 ± 4.58	5 (0–15)	5.80 ± 4.2	5 (0–13)	6.80 ± 6.10	5 (0–15)	0.684
	**Group C**		**Men**		**Women**		
	Mean ± SD	Median (IR)	Mean ± SD	Median (IR)	Mean ± SD	Median (IR)	*p*-Value
Age	56.87 ± 11.74	60.50 (35–79)	55.22 ± 13.00	55 (53–70)	59.41 ± 9.57	61 (45–72)	0.34
PWT	4.70 ± 4.46	3 (80–15)	3.77 ± 3.84	5 (0–15)	6.08 ± 5.10	5 (0–13)	0.169

PWT: prosthesis wearing time (measured in years).

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
