# Peer review of "Active Tactile Sensibility in Implant Prosthesis vs. Complete Dentures: A Psychophysical Study"

_jcm, 2022, doi:10.3390/jcm11226819_

Round 1

Reviewer 1 Report (Previous Reviewer 2)

The manuscript has been improved

Author Response

Comments and Suggestions for Authors

The manuscript has been improved

There are no considerations as reviewer 1 did not make any suggestion to improve this manuscript. Minor language modifications have been performed as this manuscript was reviewed by a native English speaker.

Reviewer 2 Report (Previous Reviewer 4)

Dear authors, I appreciate the efforts you have taken to revise your manuscript, I have few more queries- how you decided on sample size?

describe your sample size in methodology and group division with a number of samples in each group in methodology.

The discussion needs more modifications- please describe whether you accepted or rejected your hypothesis.

Also, I recommend removing the words - I, WE...these are not required in the manuscript...keep it generalised - like- in this study...or it is revealed..."we thought..." ....not scientific....

Author Response

RESPONSE TO REVIEWER 2

Comments and Suggestions for Authors

Dear authors, I appreciate the efforts you have taken to revise your manuscript, I have few more queries- how you decided on sample size?

Sample size was influenced by the number of patients wearing complete dentures nowadays. It has been very difficult to find complete denture wearers in both arches who were able to understand the procedure, as older patients had trouble collaborating in the study. Beside this there are more and more patients rehabilitated with dental implants and less with complete dentures. In order to balance the number of patients included in each group, complete denture wearers were the reference to set the sample size.

describe your sample size in methodology and group division with a number of samples in each group in methodology.

Now, sample size is described in methodology as well as number of samples of each group

The discussion needs more modifications- please describe whether you accepted or rejected your hypothesis.

The acceptance of the hypothesis has been included in discussion section.

Also, I recommend removing the words - I, WE...these are not required in the manuscript...keep it generalised - like- in this study...or it is revealed..."we thought..." ....not scientific....

These words have been removed in order to improve the manuscript. Also, the manuscript has been reviewed by a native English speaker to solve mistakes.

Reviewer 3 Report (Previous Reviewer 1)

Well, modifications were done to improve the readability of the manuscript. However few more points need to be stressed upon

1.       Tense should be corrected from future to past tense as the study has already been performed.

2.       Is the study one of its kind? Any other previous comparative study on the same topic? Their inference. need for study?

3.       Grammatical corrections need to be corrected. Sentence formation should be checked

4.       It is better in the scientific literature that we don’t mention “WE”

5.       Can more information be provided on the implant rehabilitations received by the patients like implant-supported overdentures or full mouth fixed implant-supported prosthesis, how many implants in an arch or in which arch (or should be mentioned in the limitation)

6.       Table 3 labeling should be corrected

7.       Can the mean age of patients be 5.80 in group B global?

8.       What do you mean by “ global”

9.       Can some mean and standard deviation be exactly similar between group b and group c values? Can prosthesis wearing time be exactly similar in two different groups?

10.   Did the references mentioned in the discussion 25,26,47-54 compared the implants and Complete denture prosthesis for active tactile sensibility?

11.   As a recommendation before also, it will be wise to show some clinical figures for better understanding.

12.   In conclusion are you sure to conclude that the improved tactile sensibility is because of osseoperception phenomenon.?

Author Response

RESPONSE TO REVIEWER 3

Comments and Suggestions for Authors

Well, modifications were done to improve the readability of the manuscript. However few more points need to be stressed upon

  1. Tense should be corrected from future to past tense as the study has already been performed.

Future tenses have been changed as indicated

  1. Is the study one of its kind? Any other previous comparative study on the same topic? Their inference. need for study?

 There are only 2 studies measuring and comparing active tactile sensibility in complete dentures and implants (Batista, 2008 and Shala, 2017) and one of them is too old. Our work team performed a complete review about psychophysical studies to renovate old protocols by using new procedures implemented in those articles that referred to tactile sensibility; as well as including all the information about osseoperception discovered during last years.

  1. Grammatical corrections need to be corrected. Sentence formation should be checked

Grammar has been checked by a native speaker in order to improve the manuscript.

  1. It is better in the scientific literature that we don’t mention “WE”

It has been removed and changed to more suitable words.

  1. Can more information be provided on the implant rehabilitations received by the patients like implant-supported overdentures or full mouth fixed implant-supported prosthesis, how many implants in an arch or in which arch (or should be mentioned in the limitation).

We included every implant prosthesis in the same group so we add this consideration as a limitation of this study. These characteristics are supposed to be included in further investigations about tactile sensibility.

  1. Table 3 labeling should be corrected

It has been corrected

  1. Can the mean age of patients be 5.80 in group B global?

There was a mistake in this figure. Mean age is 59.8

  1. What do you mean by “ global”

It refers to the results of the whole group, it has been removed to avoid confusion

  1. Can some mean and standard deviation be exactly similar between group b and group c values? Can prosthesis wearing time be exactly similar in two different groups?

Sorry for this mistake, it has been solved. Results from group b were also copied in group c by mistake

  1. Did the references mentioned in the discussion 25,26,47-54 compared the implants and Complete denture prosthesis for active tactile sensibility?

These references refer to those who claim that implants present an increased sensibility, since they led to confusion, have been modified to those that refer to comparison between complete dentures, exclusively.

  1. As a recommendation before also, it will be wise to show some clinical figures for better understanding.

  1. In conclusion are you sure to conclude that the improved tactile sensibility is because of osseoperception phenomenon.?

It is true that this phenomenon is not well understood yet, but there is enough information that suggest its relationship with an increased sensibility in dental implant rehabilitations. In this way, these results are explained and related to osseoperception.

This manuscript is a resubmission of an earlier submission. The following is a list of the peer review reports and author responses from that submission.

Round 1

Reviewer 1 Report

Well, thought topic for research. Overall planning and execution is commendable. However, the details of the methodology used can be improved and can be substantiated by providing a few figures to make it more clear for the readers.

Reviewer 2 Report

In this study the authors proposed to measure the sensibility in different prosthetic situations by performing a psycho-physical investigation. The patients weredivided in three groups: complete dentures in both arches,  implant prosthesis on both arches and finally implant prosthesis with natural dentition as antagonist.

The originality of this study is limited and it brings limited value as a clinical study.

Please see the enclosed PDF for details.

Reviewer 3 Report

Thank you for your interesting study. The study design looks simple and straightforward.

I believe the information would deliver worthy information to clinicians. However, there are several minor issues that should be addressed in the tables. Please find the attached file with comments.

And it is important to understand osseoperception phenomenon for this study. It would be nicer to describe more details about that in the discussion.

Reviewer 4 Report

Dear Authors,

The title of the study is very interesting and it attracted me for reading it in detail, but the manuscript lacks clarity on the study design.

Poor English, use of future tense and inappropriate sentences in materials and methods - reduces the quality of the manuscript.

what was your hypothesis? Aim is not written properly.

The methodology is not clear at all. how you had checked the subjects for the mentioned inclusion and exclusion criteria. 

"2.2. Inclusion Criteria- Only adults and healthy patients, rehabilitated with complete dentures or implant prosthesis with a proper adjustment, will be included in this investigation. 

Exclusion Criteria. 121 Patients with maladjusted prosthesis or patients suffering from temporomandibular 122 disorders will not be included in this investigation, as well as patients unable to under- 123 stand how this procedure works. Also, patients that do not pass the placebo test will be 124 excluded from the study."

ethical clearance description is missing. 

source of patient selection missing

description of variables assessed is missing - "Variables 4 different variables will be measured: type of prosthetic situation, age, gender and 127 prostheses wearing time."

results - Mean threshold value is in microns - how you got this value - description is missing

table 3,4,5 should be merged - from where you got global stats ??? reference  

The discussion is not written properly. you should compare your results in discuss it in this section.

overall manuscript lacks proper scientific presentation and requires Extensive editing of English language and style.

Hopefully, these points will help you in improving your manuscript writing.